# Deep Learning Model Based on Contrast-Enhanced Computed Tomography Imaging to Predict Postoperative Early Recurrence after the Curative Resection of a Solitary Hepatocellular Carcinoma

**DOI:** 10.3390/cancers15072140

**Published:** 2023-04-04

**Authors:** Masahiko Kinoshita, Daiju Ueda, Toshimasa Matsumoto, Hiroji Shinkawa, Akira Yamamoto, Masatsugu Shiba, Takuma Okada, Naoki Tani, Shogo Tanaka, Kenjiro Kimura, Go Ohira, Kohei Nishio, Jun Tauchi, Shoji Kubo, Takeaki Ishizawa

**Affiliations:** 1Department of Hepato-Biliary-Pancreatic Surgery, Osaka Metropolitan University Graduate School of Medicine, 1-4-3 Asahimachi, Abeno-ku, Osaka 545-8585, Japan; 2Smart Life Science Lab, Center for Health Science Innovation, Osaka Metropolitan University, 1-4-3 Asahimachi, Abeno-ku, Osaka 545-8585, Japan; 3Department of Diagnostic and Interventional Radiology, Osaka Metropolitan University Graduate School of Medicine, 1-4-3 Asahimachi, Abeno-ku, Osaka 545-8585, Japan; 4Department of Biofunctional Analysis, Graduate School of medicine, Osaka Metropolitan University, 1-4-3 Asahimachi, Abeno-ku, Osaka 545-8585, Japan

**Keywords:** hepatocellular carcinoma, deep learning, hepatectomy

## Abstract

**Simple Summary:**

Patients with postoperative early recurrence of hepatocellular carcinoma within 2 years are at high risk for poor prognosis, and identifying high-risk patients with postoperative early recurrence is becoming increasingly important in the clinical practice for hepatocellular carcinoma. However, preoperatively predicting the early recurrence remains difficult. Thus, we developed a deep learning model that accurately predicts early postoperative hepatocellular carcinoma recurrence; in addition, the contrast-enhanced computed tomography imaging analysis was the most important factor to predict early hepatocellular carcinoma recurrence in clinical variables of the current deep learning model. Guiding the treatment strategy for patients with hepatocellular carcinoma may be possible using contrast-enhanced computed tomography images by utilizing the deep learning method.

**Abstract:**

We aimed to develop the deep learning (DL) predictive model for postoperative early recurrence (within 2 years) of hepatocellular carcinoma (HCC) based on contrast-enhanced computed tomography (CECT) imaging. This study included 543 patients who underwent initial hepatectomy for HCC and were randomly classified into training, validation, and test datasets at a ratio of 8:1:1. Several clinical variables and arterial CECT images were used to create predictive models for early recurrence. Artificial intelligence models were implemented using convolutional neural networks and multilayer perceptron as a classifier. Furthermore, the Youden index was used to discriminate between high- and low-risk groups. The importance values of each explanatory variable for early recurrence were calculated using permutation importance. The DL predictive model for postoperative early recurrence was developed with the area under the curve values of 0.71 (test datasets) and 0.73 (validation datasets). Postoperative early recurrence incidences in the high- and low-risk groups were 73% and 30%, respectively (*p* = 0.0057). Permutation importance demonstrated that among the explanatory variables, the variable with the highest importance value was CECT imaging analysis. We developed a DL model to predict postoperative early HCC recurrence. DL-based analysis is effective for determining the treatment strategies in patients with HCC.

## 1. Introduction

Liver resection is a curative hepatocellular carcinoma (HCC) treatment, but the recurrence rate of HCC has been reported to be up to 70%, 5 years after resection [1,2]. HCC recurrence types are divided into early and late recurrences, and early recurrence is considered to result from a hematogenous tumor that has spread from the primary tumor. Patients with postoperative early recurrence within 2 years postoperative are at high risk for poor prognosis [3,4]. Several predictive factors for early recurrence have been identified in blood tests and pathological findings, such as serum α-fetoprotein (AFP) level, tumor size, multiple tumors, microvascular invasion, and histologic grade [2,5]. However, preoperatively predicting the early recurrence remains difficult. Recently, systemic therapy using molecularly targeted agents or immune checkpoint inhibitors has provided long-term survival in patients with HCC [6], and patients at high risk of early recurrence are potential candidates for clinical trials of adjuvant systemic therapies [7]. Thus, identifying high-risk patients with postoperative early recurrence is becoming increasingly important in the clinical practice for HCC.

Contrast-enhanced computed tomography (CECT) imaging is a very useful tool in HCC diagnosis, and its dynamic and structural information can preoperatively predict tumor subtypes and malignant potential [8]. Previous reports indicated that preoperative CECT findings could predict poor postoperative outcomes in patients with HCC [8,9]. However, diagnostic imaging assessment by clinicians may often be subjective, and it may be one of the limiting factors in the precise and personalized HCC treatment. No model for predicting postoperative early recurrence of HCC using preoperative imaging has been established.

Deep learning (DL) is emerging as an attractive technology for mining latent image features based on complex artificial neural network architecture [10,11]. The deep convolutional neural networks (CNNs) of DL are commonly utilized in image recognition, and can automatically extract and learn deep features of input data through several consecutive filters without the need for handcrafted design [12]. Several studies based on DL models have reported in the field of HCC, and models that accurately predict, such as HCC diagnosis, pathological vascular invasion prediction, and long-term prognosis, using imaging findings or histopathological specimens, have been published so far [13,14,15,16]. However, only a few predictive models for early HCC recurrence using the DL method were extracted from image findings [17].

This study aimed to establish a predictive DL model for postoperative early HCC recurrence based on CECT imaging.

## 2. Materials and Methods

This retrospective study followed the ethical guidelines of the Declaration of Helsinki. Approval was obtained from the ethics committee of Osaka Metropolitan University (No. 3166). All participants provided written informed consent.

### 2.1. Study Cohort

This study included 606 patients who underwent initial hepatectomy for solitary HCC as curative (R0) resections at the Department of Hepato-Biliary-Pancreatic Surgery, Osaka Metropolitan University from January 2007 to December 2019. Patients with undetermined suspicious lesions, such as high-grade dysplastic nodules with early washout on CECT, were excluded from the study after retrospectively reviewing the images. In addition, this study excluded 63 patients who were unavailable for preoperative CECT evaluation and included 543 patients in the study cohort (Figure 1).

### 2.2. Patient and Public Involvement

The present study did not involve any patient or public partnership.

### 2.3. Postoperative Follow-Up and Recurrence Diagnosis

Patients were followed up once every 3 months postoperatively. At each follow-up visit, the routine examination included the HCC-specific tumor marker measurement. Additionally, ultrasonography, CECT, or magnetic resonance imaging (MRI) was conducted. Postoperative recurrence was diagnosed based on increased tumor markers that declined to normal range and evidence regarding new extrahepatic or intrahepatic lesions. The preoperative images were reviewed again to determine if the extrahepatic or intrahepatic lesions were new. The patients with no suspicious lesions at the same site before surgery were only included in this cohort as patients with “postoperative recurrence after curative resection”.

### 2.4. Data Partition

Patients were randomly divided into training, validation, and test datasets based on random numbers using Python to achieve a ratio of 8:1:1, which included 434, 54, and 55 patients, respectively. There was no overlap of patients among the respective datasets (Figure 1).

### 2.5. Image Processing

All images were augmented using random rotation of −0.1–0.1 radians, with a random shift of 10%, a brightness range of 10%, and reflected horizontally. We determined the most suitable hyperparameters—optimizer, learning rate, image size, input channels, batch size, and global pooling strategy—using a grid search. Stochastic gradient descent (0.05 to 0.001 for learning rate), Adam (default parameters), and Adagrad (default parameters) were used as optimizers. Search ranges were 256, 320, and 512 pixels for image size. The longer side of the image was downscaled to the size while maintaining the aspect ratio, then the width along the shorter side was padded black to the selected size. For batch size, search ranges were 16 to 64.

### 2.6. Model Development and Evaluation

Several clinical variables and arterial preoperative CECT imaging phases [8,18] were used to create the predictive model for early HCC recurrence, including sex, age, serum alanine aminotransferase (ALT), and alpha-fetoprotein (AFP) levels, Child–Pugh classification, and platelet counts. This study defined postoperative early recurrence as intra- or extrahepatic recurrence within 2 years postoperative.

The artificial intelligence (AI) models were implemented using both CNNs and a multilayer perceptron (MLP) as a classifier. The CNN was prepared with ResNet50 [19], InceptionV3 [20], and DenseNet121 architectures [21], and the obtained CNN output using images was concatenated with the obtained MLP output using clinical data variables. These were fed into four fully connected layers, and then a final output was obtained using cross-entropy loss. The output classifies the recurrence of liver cancer.

All models were developed using the PyTorch framework [22]. Each model was trained from scratch with the training dataset and tuned with the validation dataset. The performance of each model was assessed using the independent test dataset. Further model development details are available in Figure 2, and the source code is available online “https://github.com/lc-recurrence/ (accessed on 31 May 2022)”.

### 2.7. Saliency Maps

The CECT was acquired in the axial plane with 1.0 mm thick sections. We evaluated the arterial CECT phase, and one image that depicted the largest tumor diameter was extracted and incorporated into the DL data.

A saliency map was generated for each evaluated CECT by the mixed model to visualize the model’s focus as it classified patient prognosis [23]. A detailed explanation of the saliency map generation model is shown in Figure 3, and the source code is available online “https://github.com/lc-recurrence/ (accessed on 31 May 2022)”.

### 2.8. Permutation Importance

Importance values for each explanatory variable, including CECT in the DL model, were calculated using permutation importance [24]. Permutation feature importance is a model inspection technique that is especially useful for nonlinear or opaque estimators and is a decreased model score when a single feature value is randomly shuffled. This procedure breaks the relationship between the feature and the target, thus the drop in the model score is indicative of how much the model depends on the feature.

### 2.9. Statistical Analysis

Background characteristics and surgical outcomes were summarized as the median and interquartile range for continuous variables and frequency and percentage for categorical variables. The Kruskal–Wallis and chi-square tests were used to compare continuous and categorical variables, respectively. We applied the receiver operating characteristic (ROC) curves and their area under the curve (AUC) value to evaluate the predictive performance of differential models. The Youden index, which was calculated by ROC curves, was used to discriminate between high- and low-risk postoperative early recurrence. Statistical analyses were performed using JMP version 11.

## 3. Results

### 3.1. Patient Backgrounds and Surgical Outcomes

Table 1 shows the demographic backgrounds and surgical outcomes of the patients. The median age of the study cohort was 71 years, and 17 patients were classified into Child–Pugh classification B. Hepatitis B (HB) surface antigen and hepatitis C virus (HCV) antibodies were observed in 94 and 279 patients, respectively. HBV-DNA was less sensitive to detection in 39 patients, and sustained viral reaction to HCV was achieved by interferon or direct acting antivirals in 61 patients. The median tumor diameter was 3 cm. Partial liver resection, segmentectomy, sectionectomy, bisectionectomy, and trisectionectomy were performed in 329, 54, 90, 69, and 1 patient(s), respectively. Pathological liver cirrhosis and microvascular invasion were observed in 132 and 154 patients, respectively. The median observation period was 45 months.

### 3.2. Patient Characteristics in Each Dataset

Patient characteristics, including explanatory variables, which were used to predict early HCC recurrence with the DL model in each dataset, were described in Table 2. No significant differences were found in patient characteristics among these datasets.

### 3.3. Model Development

Each model was independently developed using the training dataset and tuned with a validation dataset for 100 training epochs, and then the loss value on a separate validation dataset determined the model performance. The final optimizer for all models was Adam (learning ratio = 0.001) with a batch size of 64. The best-performing models were obtained with an image size of 512 pixels in both image-based and mixed models; DenseNet was the best-performing CNN architecture.

### 3.4. Model Evaluation

The original CECT images and their corresponding saliency heatmaps on the DL model are described in Figure 4. ROC curves in the validation and test datasets are described in Figure 5 based on the model to predict early HCC recurrence, which was established by the DL analysis in training datasets with explanatory variables, including saliency heatmaps. The AUC value of validation datasets was 0.73 and that of test datasets was 0.71. The DL model categorized 15 patients of test datasets with a high risk and 40 with a low-risk postoperative early recurrence. Postoperative early recurrence occurred in 11 (73%) patients in the high-risk and 12 (30%) in the low-risk groups (*p* = 0.0057). Preoperative clinical variables, including saliency maps, and corresponding postoperative prognosis of two patients are described in the Appendix A.

Permutation importance is described in Figure 6. A large positive value indicates that the feature is very relevant in detecting positive output. The color bar shows the feature value impact on model output (early HCC recurrence). Figure 6 demonstrates that the variable with the highest value was CECT imaging analysis in our present DL model to predict the early HCC recurrence.

## 4. Discussion

The present study suggested that our designed method could develop a DL model to predict early postoperative HCC recurrence, mainly based on preoperative CECT imaging. This DL model demonstrated high accuracy in predicting early recurrence with the AUC value of 0.71 in test datasets. Furthermore, the DL model was internally validated showing high accuracy with the AUC value of 0.73. To our knowledge, no report has shown the DL model predicting early HCC recurrence according to the classified features from CECT imaging through the totally CNN architecture process.

AI models, including DL, have been widely used for disease diagnosis, prognosis predictions, risk management, and clinical decision making [13,25,26]. DL was designed to mimic the neurological structure of the human brain [11]. Furthermore, DL does not require a human to present the features and is, thus, considerably different from conventional machine learning, which requires the extraction of features from images using advanced learning by humans, whereas the application of convolutional layers allows the image to be used during the learning process [10,11]. Hence, a computer can obtain hidden factors from all images through self-learning that have not been noticed by humans before and has the potential to develop completely new evaluations [27]. Gao et al. previously developed the DL-based predictive model for early postoperative HCC recurrence using MRI [17]. Two types of features from manually segmented tumor lesions were integrated, including the deep features extracted by deep CNN architecture, which recognizes and classifies image features, and the radiomics features, which were extracted from an open-source software platform, to develop the predictive model. Meanwhile, the CNN architecture in this study was formed using a simple CECT imaging slice by concatenating the MLP output using clinical variables. The current predictive model for early HCC recurrence showed highly accurate predictive ability.

Permutation importance in the present study indicated that CECT imaging analysis was of the highest importance to predict early postoperative recurrence among clinical variables. Several previous studies had indicated that CECT imaging findings, such as intratumor necrosis, intratumor hemorrhage, and vessel invasion, were related to tumor subtypes, malignancy, and HCC prognosis [8,9,28,29]. The present DL model might accurately predict early recurrence by adding image recognition methods different from the conventional diagnosis to these previously reported factors [8,9,28,29]. The DL model using CT and/or MRI images for the diagnosis, and microvascular invasion of HCC has been reported with high accuracy [13,14,16], and a few previous studies have indicated that the DL-based radiomics model can predict long-term prognosis using preoperative diagnostic imaging analyses [15,30]. Accordingly, we indicated the possibility of predicting early postoperative HCC recurrence using CECT images by utilizing DL with AI. However, the DL CNN algorithm recognition is usually considered similar to a “black box” in data analysis, and the operating procedures used by the DL models to generate radiological features are difficult to directly interpret [31,32,33]. Even herein, determining which part of the image was evaluated by DL-based image analysis was not possible. In addition, determining who is responsible when an error occurs in DL-analyzed results is an ethical concern [32]. These are significant concerns when using the DL model in clinical practice. While making satisfactory diagnoses and treatment strategy decisions based on the DL analysis results is difficult when the analysis details are unclear, the opportunity to utilize unknown or known unintended consequences from DL in clinical practice may be lost if DL is not used [32,33]. A recent study indicated that building accurate and explainable DL models can be achieved by building interpretability into machine learning models from scratch; this is important for DL-model-based analyses to gain the trust of clinicians and patients [32]. Thus, the ongoing collaboration between AI experts and clinicians is essential for resolving these problems and further developing DL [32].

Recently, systemic therapy using molecularly targeted agents and/or immune checkpoint inhibitors was shown to provide long-term survival in patients with HCC [6]. Many trials of immune checkpoint inhibitors are ongoing in the neoadjuvant setting [34]. Selecting candidates for preoperative adjuvant therapy for HCC may be possible using CECT images by utilizing DL with AI due to the high prediction accuracy of early recurrence in this DL model. Furthermore, when similar DL models are developed to predict the efficacy of nonsurgical treatment, such as response rates of this systemic therapy in the future, DL will become even more useful to decide the treatment strategy for HCC.

This study has several limitations. First, this is a single-center retrospective study with a relatively small patient number. Second, the DL model in this study included only the arterial CECT phase as a medical imaging dataset. Additionally, the CECT images were analyzed using a cross-section of the largest tumor diameter; thus, the entire liver, including the whole tumor, was not evaluated. The arterial CECT phase may be useful to predict microvascular invasion, tumor malignancies, and prognosis [8,29], and using only one slice of the arterial phase to accurately predict postoperative early recurrence is surprising and highly useful owing to the simplicity of the procedure. However, combining multiple CECT phases and slices may improve the accuracy of the predictive model, and comparisons and validation using models that include other phases may be necessary. In addition, similar studies using Gd-EOB-DTPA MRI, which probably exhibits higher accurate diagnostic capabilities, should be considered. Finally, the DL model was developed only for patients with solitary HCCs. Although the results indicate the practical utility of our model, additional studies using the Milan criteria, which are useful for determining resectability not only in HCC but also in metastatic liver cancer [35], should be conducted to improve the clinical utility of the model. After attempting to improve the accuracy with these methods, we will continue to investigate with a multicenter prospective study.

## 5. Conclusions

In conclusion, our DL model enables accurate early postoperative HCC recurrence prediction based on CECT imaging analysis. The present result strongly suggests that DL-based analysis may be effective for determining the treatment strategies in patients with HCC.

## Figures and Tables

**Figure 1 cancers-15-02140-f001:**
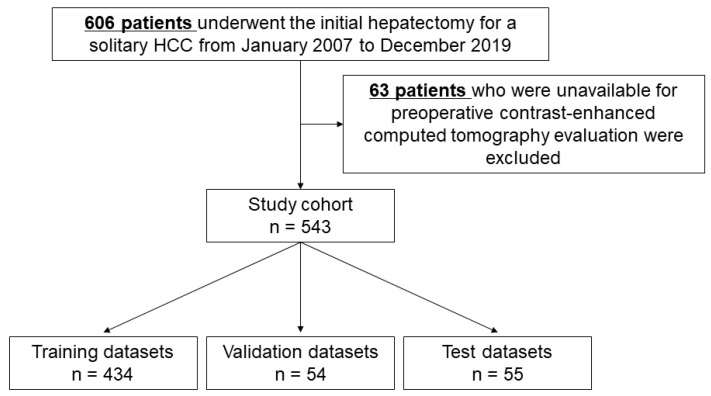
Flowchart of dataset selection.

**Figure 2 cancers-15-02140-f002:**
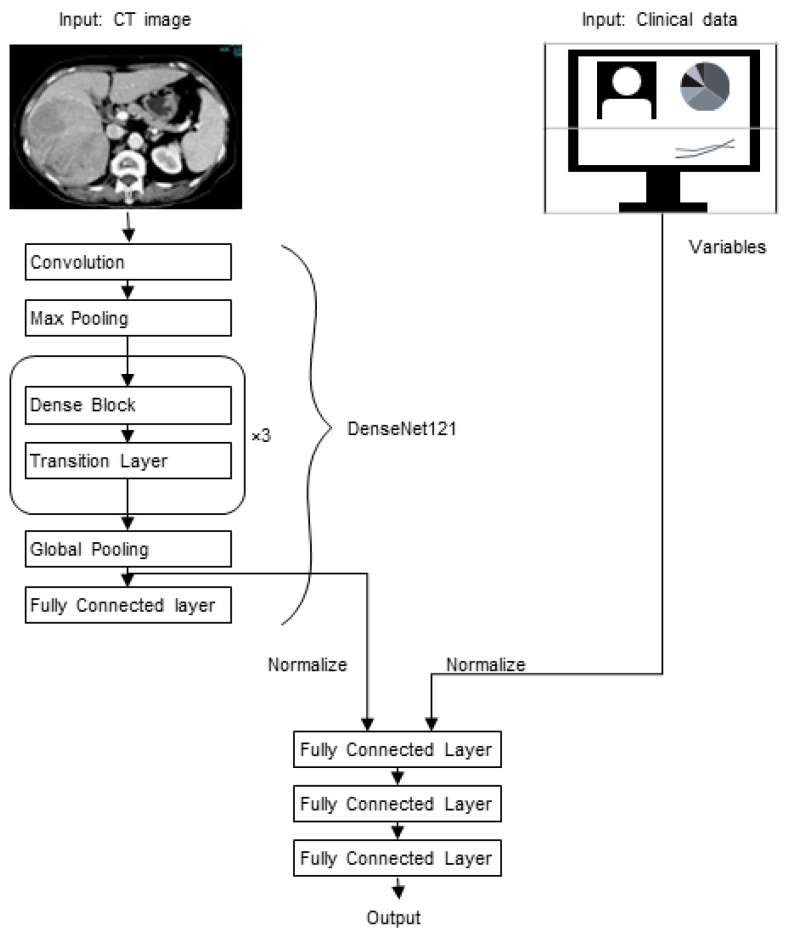
Outline of the mixed artificial intelligence model. In this model, we used one variable from convolutional neural network (CNN) output and 6 variables from clinical data. The output of CNN and clinical data were used as input for multiple perceptrons (MLP), and the output of MLP was arranged to show recurrence or not. Our trials evaluated InceptionV3, ResNet50, and DenseNet121 as feature extractors. DenseNet121 achieved the highest performance in the model, which is discussed here. First, the CT images were fed into the feature extractor that was composed of convolutional layers, wherein the images were downsampled by half each pass through the convolutional layers. The feature extractor results were connected to a classifier, in which the affine was followed by a sigmoid activation function and a binary cross-entropy loss function. This process was repeated many times with the training data to optimize parameters for a model that can determine patient prognosis when developing the model. DenseNet121 consists of five parts, including convolutional layer1 and Dense block1, 2, 3, and 4. Convolutional layer1 consists of one 7 × 7 convolutional layer. Each Dense block consists of one 1 × 1 and two 3 × 3 convolutional layers. Block1 is repeated 6 times, block2 is repeated 12 times, block3 is repeated 24 times, and block4 is repeated 16 times. A 1 × 1 convolutional layer and one max-pooling layer, collectively called the transition layer, is present at the transition between blocks 1 and 2, blocks 2 and 3, and blocks 3 and 4. FC, fully connected layer.

**Figure 3 cancers-15-02140-f003:**
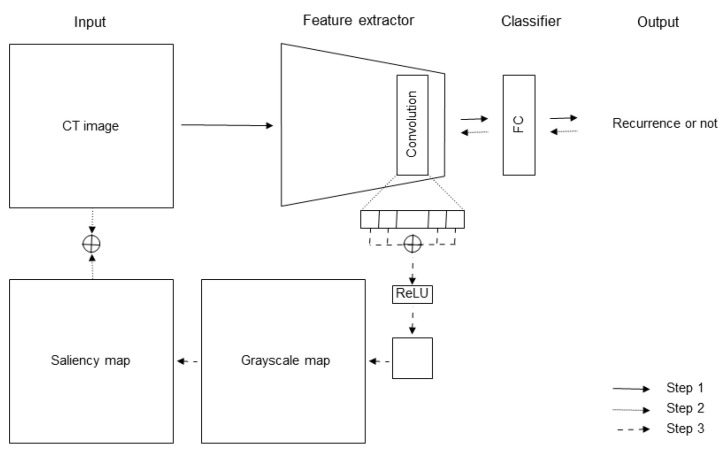
Outline of the Grad-CAM model. In a nutshell, Grad-CAM [23] works by inputting an image of the object to be visualized into a trained model and then outputting a heat map of the final convolutional layer in the model that reacts to the object. Step 1 is a forward propagation phase. A trained model is used to classify the target to be visualized. In our study, the target was a CT image from patients predicted with early postoperative recurrence. Grad-CAM visualizes features of these images to highlight the features that are most impacted by the model. The CT image is fed into the model, and the output of the final convolutional layer in the model is obtained. Step 2 is the backpropagation phase, wherein the results responding to only the class (image from patient with recurrence) to be visualized are obtained in the backpropagation. Step 3 is an image synthesis phase, wherein the synthesized image is obtained by multiplying the images created in Step 1 and the matrices in Step 2. The synthesized image is normalized after passing ReLU and finally resized to the original image size to complete the heat map creation. Finally, the heat map and original image (CT image) are concatenated. FC, fully connected layer; ReLU, rectified linear unit.

**Figure 4 cancers-15-02140-f004:**
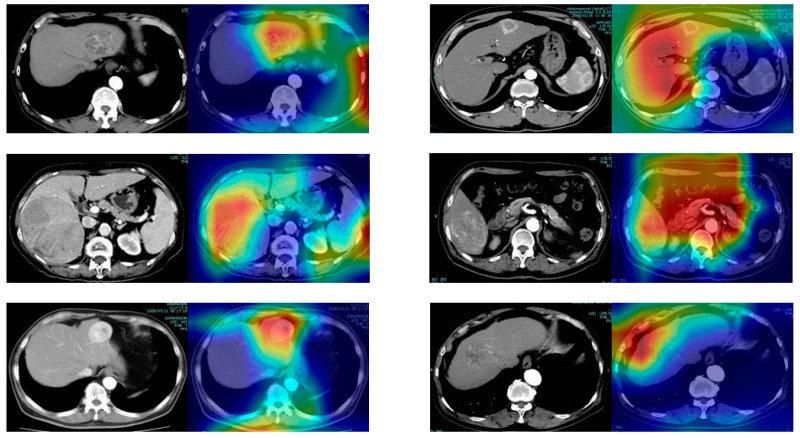
The original CECT images and their corresponding heatmaps are shown from left to right. The red color highlights the region of interest to predict early recurrence. CECT, contrast-enhanced computed tomography.

**Figure 5 cancers-15-02140-f005:**
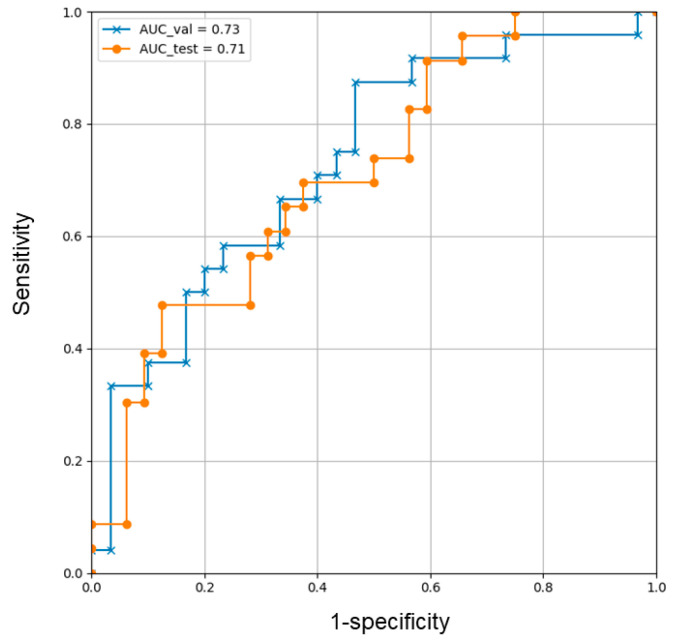
The receiver operating characteristic curves of the validation (blue) and test (orange) datasets. Area under the curves in the validation and test datasets were 0.73 and 0.71, respectively.

**Figure 6 cancers-15-02140-f006:**
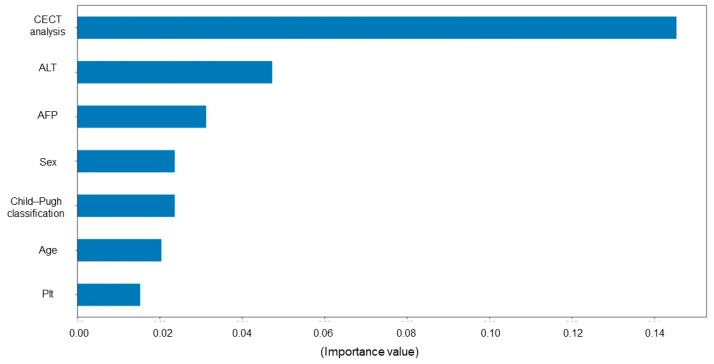
Permutation importance in each explanatory variable of our present deep learning model. A large positive value indicates the feature is very relevant to detect positive output. The color bar shows the feature value impact on early recurrence. CECT, contrast-enhanced computed tomography; ALT, alanine aminotransferase; AFP, alpha-fetoprotein; Plt, platelet counts.

**Table 1 cancers-15-02140-t001:** Background characteristics and surgical outcomes of the study cohort.

Variables	Value
Age, years, median (range)	71 (19–87)
Sex, male/female	401/136
Comorbidities, n (%)	
Diabetes mellitus	187 (34)
Hypertension	287 (53)
Dyslipidemia	107 (20)
HB surface antigen positive	94 (17)
HBV-DNA < detectable levels	39 (7.2)
HCV antibody positive	279 (51)
HCV-SVR	61 (11)
Laboratory data, median (range)	
Total bilirubin, mg/dL	0.6 (0.1–2.7)
ALT, IU/L	29 (6–270)
Albumin, g/dL	4.0 (2.3–5.1)
PT, %	94 (40–147)
Platelet count, ×10^4^/μL	15.0 (1.3–42.8)
AFP, ng/mL	9.1 (1.5–283,300)
PIVKA-II, mAU/mL	71 (2–3893,20)
Child–Pugh classification, A/B	526/17
Tumor diameter, cm, median (range)	3 (0.7–19.5)
Hepatectomy procedures	
Partial resection	329 (61)
Segmentectomy	54 (9.9)
Sectionectomy	90 (17)
Bisectionnectomy	69 (13)
Trisectionnectomy	1 (0.2)
Operative time, min, median (range)	278 (75–776)
Intraoperative blood loss, g, median (range)	280 (5–7460)
Postoperative complication *, *n* (%)	71 (13)
Liver cirrhosis, *n* (%)	132 (24)
Microvascular invasion, *n* (%)	154 (28)
Recurrence-free survival, months, median (range)	19 (1–170)
Early recurrence within 2 years, *n* (%)	220 (41)
Intrahepatic recurrence	195 (36)
Extrahepatic recurrence	31 (5.7)
Observed period, months, median (range)	46 (1–170)

HBV, hepatitis B virus; HCV, hepatitis C virus; SVR, sustained viral reaction; ALT, alanine aminotransferase; PT, prothrombin activation; AFP, alpha-fetoprotein; PIVKA-II, protein induced by vitamin K absence or antagonist-II. * Clavien–Dindo IIIa or greater.

**Table 2 cancers-15-02140-t002:** Patient’s characteristics in each dataset.

Variables	Training Datasets (*n* = 434)	Validation Datasets (*n* = 54)	Test Datasets (*n* = 55)	*p*-Value
Sex, male/female	331/103	37/17	38/17	0.26
Age, years	71 (31–87)	67 (19–84)	70 (38–82)	0.11
ALT, IU/L	29 (8–270)	28 (11–162)	28 (6–126)	0.78
AFP, ng/mL	8.9 (1.5–283,300)	10 (2.3–109,402)	9.4 (1.9–67,700)	0.87
PIVKA-II, mAU/mL	64 (2–389,320)	104 (13–57,202)	117 (9–228,533)	0.62
Child–Pugh classification B, *n* (%)	14 (3.2)	2 (3.7)	1 (1.8)	0.80
Platelet count, ×10^4^/μL	15.3 (1.3–42.8)	14 (5.2–37.1)	15.2 (2.2–30.3)	0.91
Tumor diameter, cm	3 (0.9–15.0)	3 (0.7–19.5)	3.3 (0.9–18.2)	0.69
≥Bisectionectomy, *n* (%)	56 (13)	6 (11)	8 (15)	0.87
Operative time, min	272 (93–643)	310 (75–776)	300 (127–563)	0.17
Intraoperative blood loss, g	275 (5–7460)	275 (5–3750)	360 (5–6265)	0.23
Liver cirrhosis, *n* (%)	100 (23)	17 (32)	15 (27)	0.34
Microvascular invasion, *n* (%)	126 (29)	17 (32)	11 (20)	0.33
Recurrence-free survival, months	20 (1–161)	18 (1–113)	17 (1–170)	0.45
Early recurrence within 2 years, *n* (%)	173 (40)	24 (44)	23 (42)	0.79

median (range) ALT, alanine aminotransferase; AFP, alpha-fetoprotein.

## Data Availability

All data generated or analyzed during this study are included in this article and its Appendix A. Further enquiries can be directed to the corresponding author.

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
