# Peer review of "Deep Learning Model Based on Contrast-Enhanced Computed Tomography Imaging to Predict Postoperative Early Recurrence after the Curative Resection of a Solitary Hepatocellular Carcinoma"

_cancers, 2023, doi:10.3390/cancers15072140_

Round 1

Reviewer 1 Report

Within 2 years postoperatively, recurrence of hepatocellular carcinoma (HCC) patients have poor prognosis and are at high risk. Accordingly, identifying these high-risk patients is critical for improving outcomes for this disease. Unfortunately, preoperative prediction of early recurrence remains a significant challenge. Here, the authors developed a deep-learning (DL) predictive model for postoperative early recurrence of HCC based on contrast-enhanced computed tomography (CECT) imaging. 543 patients who underwent initial hepatectomy for HCC were included and randomly classified into training, validation, and test datasets. Clinical variables and arterial CECT images were used to produce prognostic models for early recurrence. Artificial intelligence models were used as was the Youden index to distinguish high vs low risk groups. This DL predictive model was able to predict postoperative early HCC recurrence accurately as described by the authors. Overall, the the manuscript concludes that DL-based analysis can be effective for determining the treatment strategies in patients with HCC. The study is well designed and executed. Data is clearly presented, with flow charts illustrating methods. Overall, this is a nice study that will be of interest to the field.

Reviewer 2 Report

In their single center retrospective studythe authors aimed to establish a predictive DL model for postoperative early recurrence of resected HCC based on CECT imaging. There is indeed a significant need for predictors of recurrence of HCC following resection especially given the high recurrence rate even after R0 resection. Nevertheless, the study suffers from singificant problems including: a) The first part of the methods section needs to be moved to the results, b) Only a very small number of patient and tumor characteristics are presented ie. R0% resection, MVI, cirrhosis% etc c) Clearly this model refers to a very specific subset of patients with low AFP, questionable cirrhosis status, unknown MVI status and so on. While this work is partly interesting it is signficantly flawed.

Reviewer 3 Report

This is a good trial but the characteristics should be reevaluated:

The data is too clean to represent the whole clinical condition and with some discrepancies,

such as

First: Table 1

(1)  Age ranges from 65-74, why there is no younger 30 to 60 y/o patients

(2)  Total Bilirubin ranges from 0.5 to 0.9, why are you excluding 1-1.9 or more

(3)  ALT ranges from 19 to 43, why there was no active hepatitis

(4)  No mention of antiviral therapy use which could be another predictor

(5)  AFP ranges from 4.2 to 58.3, but the supplementary file(s) patient 1’s AFP 327

(6)  as reimbursed in Japan, why is there no PIVKA-II nor ALP-L3

(7)  Tumor size ranges from 2 to 4.9, but the tumor size in figure 2 and figure 4 should be massive type and larger than 5 cm

(8)  How about liver cirrhosis factor, as this is surgical pathology that should be reported

Second: figure 4

Why the red heatmap color outside the tumor has the perdition power (right middle picture, patient 2)? and is the background liver also predictive in the right upper picture? (patient 1, are there any clues that there are many hypo-vascular nodules ?)

Third: line 125: why did you only choose arterial imaging? the early recurrence often happened from the undetermined nodule without early enhancement but early washout, which was often classified as a high-grade dysplastic nodule, I am disappointed that you did not provide the information to help the reader to improve their clinical knowledge.

Fourth: when you found the patients got recurrences by CT or MRI, did you recheck the pre-operation images that any clues were there? which is the most important to us to learn! and besides, can you tell us the choice of pre-operation dynamic image study is CT, which is better than MRI in some points?

I suggest you revise the key points that we want to know.

Reviewer 4 Report

This is an interesting article describing a deep learning (DL) predictive model based on a single CT scan image with arterial enhancement to predict recurrence after resection for a solitary hepatocellular cancer less than or equal to 4.9cm.  I only have a few suggestions to help make this study more accessible to our readers.

1. Although the authors discuss the concerns of the Black Box effect, the inability/difficulty in explaining the methodology devised by the deep learning architecture, I believe this issue needs to be expanded upon.

Please see this review article : Elyan E, Vuttipittayamongkol P, Johnston P, Martin K, McPherson K, Moreno-García CF, Jayne C, Mostafa Kamal Sarker M. Computer vision and machine learning for medical image analysis: recent advances, challenges, and way forward. Art Int Surg 2022;2:24-45. http://dx.doi.org/10.20517/ais.2021.15

In this article, the last paragraph discusses the dilemma that may arise as patients and certifying agencies refuse to allow clinical decisions based on AI.

This issue is further elaborated on in this article : Decker JM, Sesti J, Turner AL, Paul S. The cassandra paradox: looking into the crystal Ball of radiomics in thoracic surgery. Art Int Surg 2022;2:57-63. http://dx.doi.org/10.20517/ais.2022.05

How will patients respond when they are told that a lesion that is not currently cancer, but has a high likelihood of turning into cancer based on a DL algorithm that we don't fully understand? Will they opt for treatment? Should we propose treatment?

Additionally, what happens when we tell patients that a HCC that was resected has a high likelihood of recurrence, and as a result, they will need additional treatments and possibly chemotherapy or immunotherapy (which have there own risks) because of DL that we don't fully understand?

2. Medical doctors without experience in AI may have a difficult time understanding the differences between Machine Learning and Deep Learning and the ethical issues regarding who should be held responsible for a clinical error that results from an error of a DL predicting model such as this. This issue is described nicely in a recent Review Article on the topic.

Taher H, Grasso V, Tawfik S, Gumbs A. The challenges of deep learning in artificial intelligence and autonomous actions in surgery: a literature review. Art Int Surg 2022;2:144-58. http://dx.doi.org/10.20517/ais.2022.11

3. Lastly, I see that only solitary HCC less than 5cm were included in this study. Commonly, HCC's meeting the Milan Criteria are used for determining resectability and other treatment modalities. In fact, the Milan Criteria are now even being used to determine resectability for colorectal liver metastases.

Gumbs, A.A.; Lorenz, E.; Tsai, T.-J.; Starker, L.; Flanagan, J.; Benedetti Cacciaguerra, A.; Yu, N.J.; Bajul, M.; Chouillard, E.; Croner, R.; Abu Hilal, M. Study: International Multicentric Minimally Invasive Liver Resection for Colorectal Liver Metastases (SIMMILR-CRLM). Cancers 202214, 1379. https://doi.org/10.3390/cancers14061379

Do the authors have any plans to repeat the study in patients with 3 or fewer lesions less than 3cm (the rest of the Milan criteria)? They mention repeating this study with more images and the addition of the venous phase of the CT scan, however, for the practicing liver surgeon a discussion on how we can use this type of model for patients fulfilling the Milan criteria may be more practical and applicable to the daily care of patients.

Round 2

Reviewer 2 Report

The authors have made a substantial effort to improve their work. Nevertheless I remain skeptical as to whether this model was indeed based on a representative patient population. The title remains partly misleading as it does not for example state that it only focuses on patients with solitary HCC as also mentioned in their limitations.

Reviewer 3 Report

please add the PIVKIA-II to Figure 6. Permutation importance

Round 3

Reviewer 2 Report

The authors have made a significant effort to improve their work.